# In vivo *Firre* and *Dxz4* deletion elucidates roles for autosomal gene regulation

Daniel Andergassen[1], Zachary D Smith[1], Jordan P Lewandowski[1], Chiara Gerhardinger[1], Alexander Meissner[1,2]*, John L Rinn[3]*

[1]Department of Stem Cell and Regenerative Biology, Harvard University, Cambridge, United States; [2]Department of Genome Regulation, Max Planck Institute for Molecular Genetics, Berlin, Germany; [3]Department of Biochemistry, University of Colorado Boulder, Boulder, United States

**Abstract** Recent evidence has determined that the conserved X chromosome mega-structures controlled by the *Firre* and *Dxz4* loci are not required for X chromosome inactivation (XCI) in cell lines. Here, we examined the in vivo contribution of these loci by generating mice carrying a single or double deletion of *Firre* and *Dxz4*. We found that these mutants are viable, fertile and show no defect in random or imprinted XCI. However, the lack of these elements results in many dysregulated genes on autosomes in an organ-specific manner. By comparing the dysregulated genes between the single and double deletion, we identified superloop, megadomain, and *Firre* locus-dependent gene sets. The largest transcriptional effect was observed in all strains lacking the *Firre* locus, indicating that this locus is the main driver for these autosomal expression signatures. Collectively, these findings suggest that these X-linked loci are involved in autosomal gene regulation rather than XCI biology.

DOI: https://doi.org/10.7554/eLife.47214.001

**\*For correspondence:**
meissner@molgen.mpg.de (AM);
john.rinn@colorado.edu (JLR)

**Competing interests:** The authors declare that no competing interests exist.

## Introduction

In female mammals, one of the two X chromosomes is inactivated to compensate for gene dosage between males and females (*Lyon, 1961*), a process termed X-chromosome inactivation (XCI). In mice, there are two successive waves of XCI: imprinted and random. Imprinted XCI starts early in development by specifically inactivating the paternal X chromosome (Xp) (*Kay et al., 1994*; *Okamoto et al., 2004*). While the Xp remains silenced in extra-embryonic lineages (*Takagi and Sasaki, 1975*), it is reactivated in the embryo during implantation, followed by random XCI (*Mak et al., 2004*). After this decision has been made, the inactive X (Xi) chromosome is epigenetically maintained throughout cell division as a compact chromatin structure known as the 'Barr body' (*Barr and Bertram, 1949*).

Recent studies using chromosome conformation capture (3C) based methods have identified that the Xi folds into two megadomains and forms a network of long-range interactions termed superloops that are directed by the non-coding loci *Firre* and *Dxz4* (*Rao et al., 2014*; *Horakova et al., 2012*; *Deng et al., 2015*). Deletion of *Dxz4* in cell lines leads to the loss of both megadomain and superloop (*Giorgetti et al., 2016*; *Bonora et al., 2018*; *Froberg et al., 2018*; *Darrow et al., 2016*), while deletion of *Firre* alone disrupts the superloop but has no impact on megadomain formation (*Froberg et al., 2018*; *Barutcu et al., 2018*). Moreover, the *Firre* locus is transcribed into the long non-coding RNA (lncRNA) *Firre* that escapes XCI and plays a role in nuclear organization (*Hacisuleyman et al., 2014*; *Andergassen et al., 2017*; *Berletch et al., 2015*; *Bergmann et al., 2015*; *Yang et al., 2015*). Recent studies in human and mouse cell line models of random XCI, found that deletion of these elements have minimal impact on X chromosome biology beyond the loss of these structures, though the phenotypic consequences of their deletion throughout mammalian

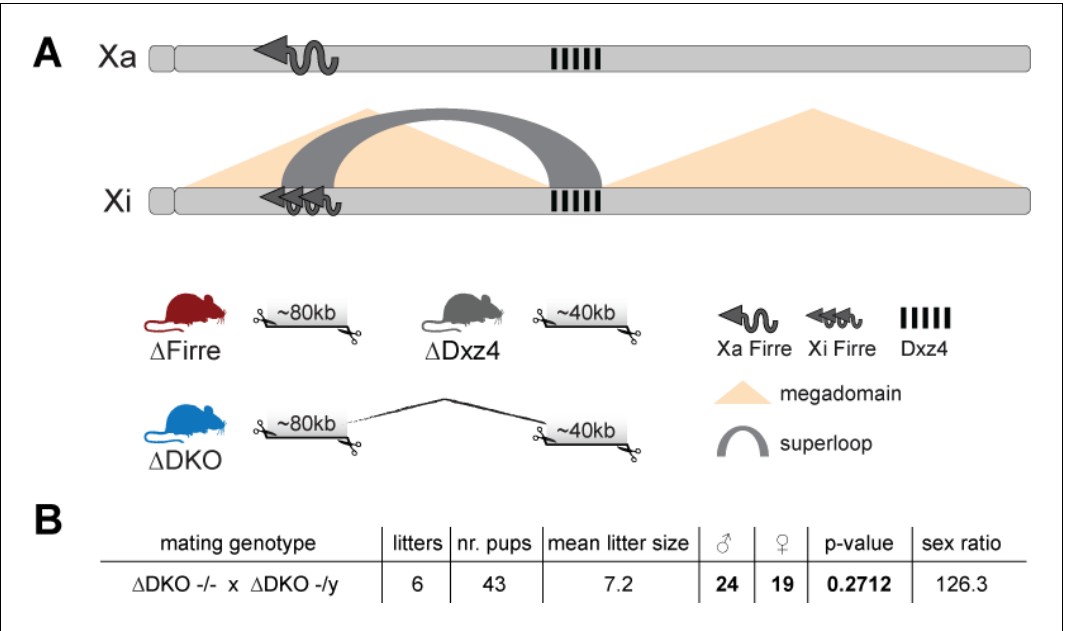

**Figure 1.** Mice carrying a single or double deletion of *Firre* and *Dxz4* are viable and fertile and show the expected litter sizes and sex ratios. (**A**) Schematic representation of the active (Xa) and inactive X (Xi) chromosomes. The deleted loci of the SKO and DKO mouse strains are indicated (*Firre* (red), *Dxz4* (gray) and DKO (blue)). The *Firre* locus escapes random XCI resulting in a full-length transcript from Xa chromosome and multiple short isoforms from Xi. (**B**) Sex type results from homozygote intercrosses. The *p*-value was calculated based on binomial distribution using R (binom.test(x = 19, n = 43, p=0.5, alternative = 'less', conf.level = 0.95)) and the sex ratio by using the following formula: number of males/number of females x 100.

DOI: https://doi.org/10.7554/eLife.47214.002

The following figure supplement is available for figure 1:

**Figure supplement 1.** Generation of mice that carry a single or double deletion of *Firre* and *Dxz4*.

DOI: https://doi.org/10.7554/eLife.47214.003

development have not been addressed (*Giorgetti et al., 2016*; *Bonora et al., 2018*; *Froberg et al., 2018*; *Darrow et al., 2016*; *Barutcu et al., 2018*).

Here, we answer this question by generating mice lacking *Firre* and *Dxz4* and performing an extensive transcriptomic analysis in embryonic, extraembryonic and adult organs. We determined that these elements are dispensable for mouse development and XCI biology. However, the absence of these loci results in organ-specific expression changes on autosomes, suggesting that these regions are involved in autosomal gene regulation rather than XCI biology.

## Results

In order to test the in vivo role of *Firre* and *Dxz4* loci both individually and in combination, we generated three knockout (KO) mouse strains: two carrying a single locus deletion (SKO) of either *Firre* (*Lewandowski et al., 2019*) or *Dxz4* and one carrying a double locus deletion (DKO) of *Firre* in conjunction with *Dxz4* (*Figure 1A*, *Figure 1—figure supplement 1A*, Materials and methods). Notably, we targeted the regions that have been previously reported to disrupt the superloop (*Barutcu et al., 2018*) and megadomain structures (*Bonora et al., 2018*). All founder mice were screened by PCR using primers that span the deleted region and identified mutants were confirmed by Sanger sequencing (*Figure 1—figure supplement 1B–C*, Materials and methods).

We found that homozygous mice of all three strains are viable and fertile, and by crossing males and females carrying a homozygous double deletion we observed the expected litter sizes and sex ratios (*Figure 1B*). To test whether the absence of the *Firre* and *Dxz4* loci has an impact on random

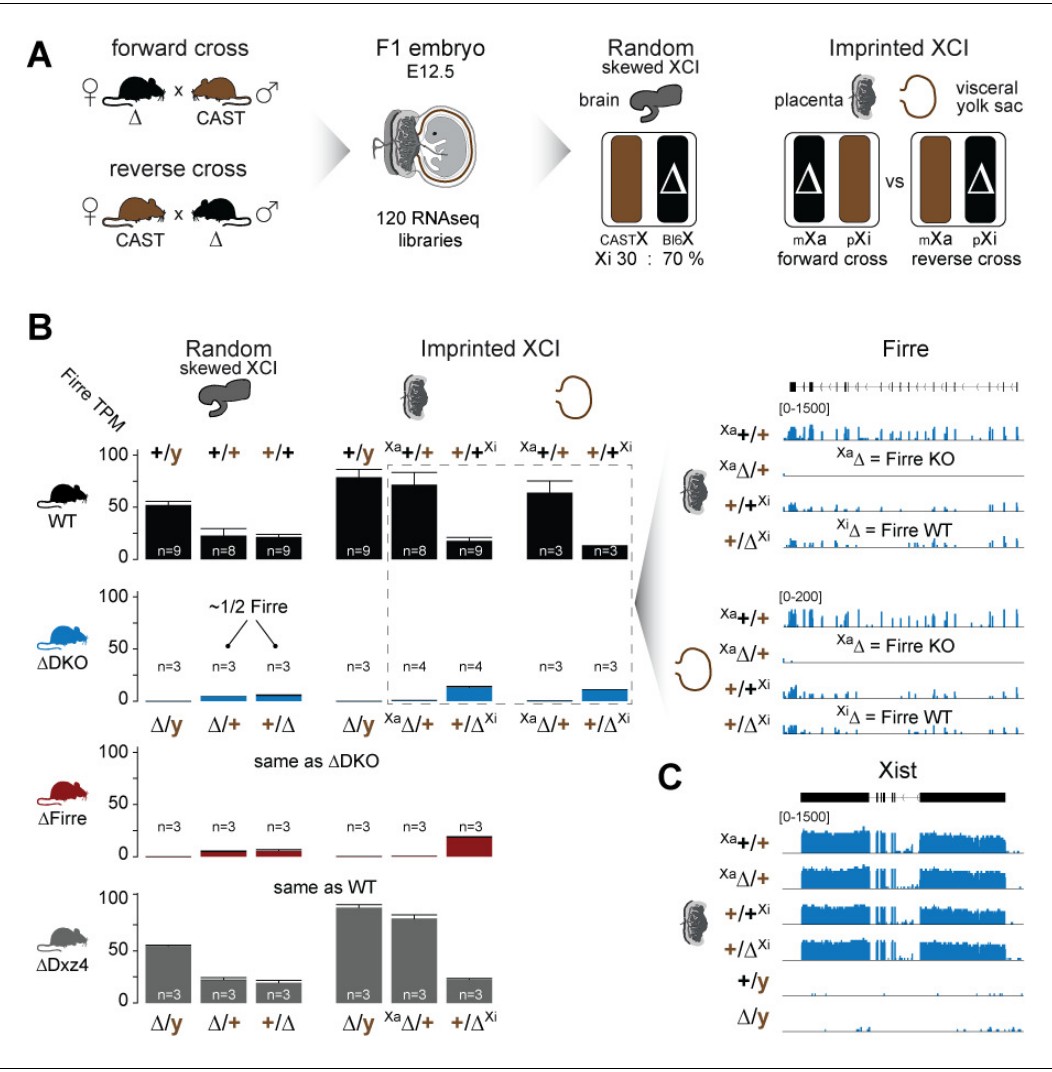

**Figure 2.** *Firre* is exclusively expressed from the maternal X chromosome in extra-embryonic tissues. (**A**) Allele-specific RNA-seq approach to test the functional impact of the maternal and paternal deletion on random and imprinted XCI using the Allelome.PRO pipeline (***Andergassen et al., 2015***). While the brain undergoes random/skewed XCI (Bl6 X chromosome inactive in 70% of cells), the placenta and visceral yolk sac undergo imprinted XCI (paternal X chromosome 100% inactive). Allele-specific analysis of the placenta and visceral yolk sac allows to distinguish the effects of maternal inheritance of the deletion (Xa, forward cross) versus paternal inheritance of the deletion (Xi, reverse cross). (**B**) *Firre* expression (mean and SD) in female and male brains, placentas and visceral yolk sacs for WT and KO mouse strain. Notably, *Firre* is approximately 4 times higher expressed from the Bl6 allele compared to the CAST allele, as observed by comparing the expression levels between the forward cross (Xa Bl6) and reverse cross (Xa CAST) in the placenta and visceral yolk sac. (**C**) *Xist* expression abundance in placenta for the *Firre-Dxz4* double KO strain.

DOI: https://doi.org/10.7554/eLife.47214.004

The following figure supplements are available for figure 2:

**Figure supplement 1.** RNA-seq quality control from F1 brains, placentas, and visceral yolk sacs.
DOI: https://doi.org/10.7554/eLife.47214.005

**Figure supplement 2.** *Firre*, *Dxz4* and *Xist* expression across tissue, sex and genotype.
DOI: https://doi.org/10.7554/eLife.47214.006

or imprinted (Xp chromosome inactive) XCI in vivo, we collected embryonic day 12.5 female F1 brains (random/skewed XCI), placentas and visceral yolk sac (imprinted XCI) from reciprocal crosses between our KO strains and *Mus musculus castaneus* (CAST), followed by RNA sequencing (RNA-

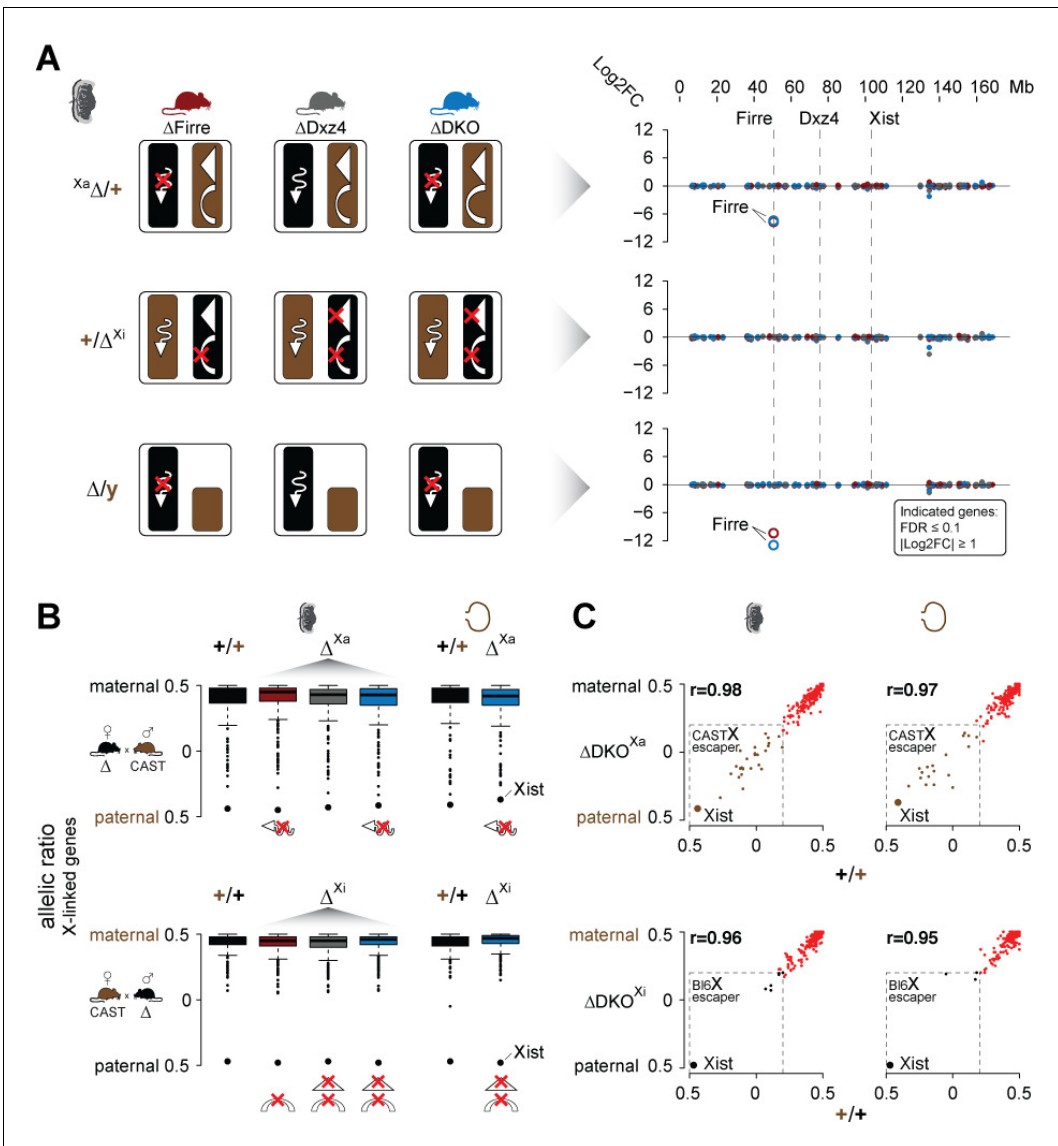

**Figure 3.** Mice carrying a single or double deletion of *Firre* and *Dxz4* undergo normal random and imprinted X chromosome inactivation (XCI). (**A**) Schematic overview showing the effect in the placenta of the deletion in females on Xa (top) or Xi (middle), and in males (bottom) for every KO strain (left). Log2FC across the X chromosome between wildtype and KO strains (right). *Firre* is the only differentially expressed gene on the X chromosome (DEseq2: FDR ≤ 0.1, |log2FC| ≥ 1). (**B**) Boxplot showing the allelic ratios for X-linked genes in the placenta and visceral yolk sac in WT and in the KO strains, for the forward cross (deletions on the maternal X = Xa) and reverse cross (deletions on the paternal X = Xi). (**C**) Scatter plot showing the allelic ratios for X-linked genes in the placenta and visceral yolk sac between WT and DKO on Xa and Xi. Pearson correlation coefficient r. Maternal ratios (red) and strain-specific escaping from the CAST (brown) and Bl6 (black) X chromosome are indicated (dashed line, escaper cutoff: allelic ratio < 0.2).

DOI: https://doi.org/10.7554/eLife.47214.007

The following figure supplements are available for figure 3:

**Figure supplement 1.** Placentas lacking *Firre* and *Dxz4* on Xa or Xi show a similar methylation levels as in wildtype.
DOI: https://doi.org/10.7554/eLife.47214.008

**Figure supplement 2.** Allelic ratio analysis from F1 brains, placentas and visceral yolk sacs.
DOI: https://doi.org/10.7554/eLife.47214.009

seq) and allele-specific analysis using the Allelome.PRO pipeline (*Andergassen et al., 2015*) (*Figure 2A*). We collected the same organs from males to test the role of these loci on the X chromosome outside of XCI biology. Unsupervised clustering of the sequenced samples confirmed the identity of the tissues (*Figure 2—figure supplement 1*). We then investigated the *Firre* expression level in *Firre* SKO and DKO female brains, and detected approximately half of the wildtype levels as expected for random XCI (*Figure 2B*, *Figure 2—figure supplement 2*). Next, we inspected the *Firre* expression level in the placenta and visceral yolk sac (extra-embryonic organs that undergo imprinted XCI) to distinguish the phenotypic consequences of deletions if they are on the Xa (maternal inheritance of the deletion) versus the Xi (paternal inheritance of the deletion) chromosome. We did not detect *Firre* expression if the deletion was inherited maternally, while we detected *Firre* wildtype expression levels in the paternal deletion, demonstrating that *Firre* is exclusively expressed from the Xa (*Figure 2B*, *Figure 2—figure supplement 2*). Thus, in contrast to previous reports that identified *Firre* as a gene that escapes XCI in cell lines that model random XCI (*Hacisuleyman et al., 2014*; *Andergassen et al., 2017*; *Berletch et al., 2015*), we found that *Firre* does not escape imprinted XCI. Remarkably, in the *Dxz4* SKO we detected a *Firre* wildtype pattern, suggesting that disruption of the superloop between the two loci or the absence of megadomains has no impact on *Firre* gene expression in either random or imprinted X inactivation (*Figure 2B*). We did not observe changes in *Xist* expression levels in the absence of these loci (*Figure 2C*).

Since *Firre* is only expressed from the Xa in the placenta and visceral yolk sac (imprinted XCI), we can use these tissues to disentangle the functional roles of *Firre* RNA from megadomain and superloop structures that only exist on the Xi. We hypothesized that: (1) females and males carrying a maternal *Firre* single or *Firre-Dxz4* double deletion (deletion on Xa) lack the *Firre* lncRNA, (2) females carrying a paternal *Firre* deletion (deletion on Xi) lack the superloop and (3) females carrying a paternal *Dxz4* single or *Firre-Dxz4* double deletion (deletion on Xi) lack both the superloop and megadomains (*Figure 3A* left panel). To identify dysregulated X-linked genes for each possible combination of the deletion, we performed differential expression analysis and found that the only dysregulated gene on the X chromosome was the lncRNA *Firre* (FDR $\leq$ 0.1 and |log2FC| $\geq$ 1), suggesting that the mega-structures and the *Firre* lncRNA have no impact on imprinted XCI (*Figure 3A* right panel, *Supplementary file 1* sheet A-B). Notably, by using the same criteria we detected only a few autosomal genes dysregulated in the DKO that were not changed in the SKO, suggesting that the mega-structures and the lncRNA *Firre* have no impact on autosomal gene regulation in the placenta (*Supplementary file 1* sheet A-B). DNA methylation levels on CpG islands were also not affected in the absence of the mega-structures or the lncRNA *Firre*, suggesting that the epigenetic processes involved in establishing the inactive X proceed normally without either (*Figure 3—figure supplement 1*, *Supplementary file 1* sheet C).

Next, we performed an allele-specific expression analysis to test for deviations of the expected maternal ratios or a gain of gene escape in the presence of the deletions. We found that the median allelic ratio of all the X-linked genes was unchanged regardless if the deletions were on Xa or Xi (*Figure 3B*). Moreover, deletion of these loci on the Xa or Xi did not result in increased escape in the placenta or visceral yolk sac (*Figure 3C*, *Supplementary file 1* sheet D-E). Of note, allele-specific analysis revealed strain-specific escaping with a greater number of gene escape from CAST Xi compared to Bl6 Xi (*Figure 3C*, *Figure 3—figure supplement 2A*, *Supplementary file 1* sheet D-E). Random XCI in the brain also did not appear to be affected, since we detect the expected XCI skewing ratios in the presence of the deletions, a well-documented effect in female cells from crosses between Bl6 and CAST that results in the predominant inactivation of the Bl6 X chromosome (*Calaway et al., 2013*) (*Figure 3—figure supplement 2B* left panel). However, our DKO animals show significant skewing of the *Xist* allelic ratios (t-test, FDR-adjusted p-value=0.0108) (*Figure 3—figure supplement 2B* right panel), suggesting that the Bl6 chromosome is further biased toward silencing in mice lacking both *Firre* and *Dxz4*. In contrast, for autosomal genes we detect the expected biallelic ratios across all samples (*Figure 3—figure supplement 2C*).

To address whether the absence of both the mega-structures and *Firre* RNA has an impact on gene expression in an organ-specific manner, we generated a transcriptomic bodymap from adult females carrying a homozygous double *Firre-Dxz4* deletion (*Figure 4A*, *Figure 4—figure supplement 1A–B*). We then performed differential expression analysis of spleen, brain, kidney, heart, lung and liver and found that most of the dysregulated genes show organ-specific expression changes, primarily on autosomes (autosomes: 98.15% n = 372 chrX: 1.85% n = 7), with only a few overlapping

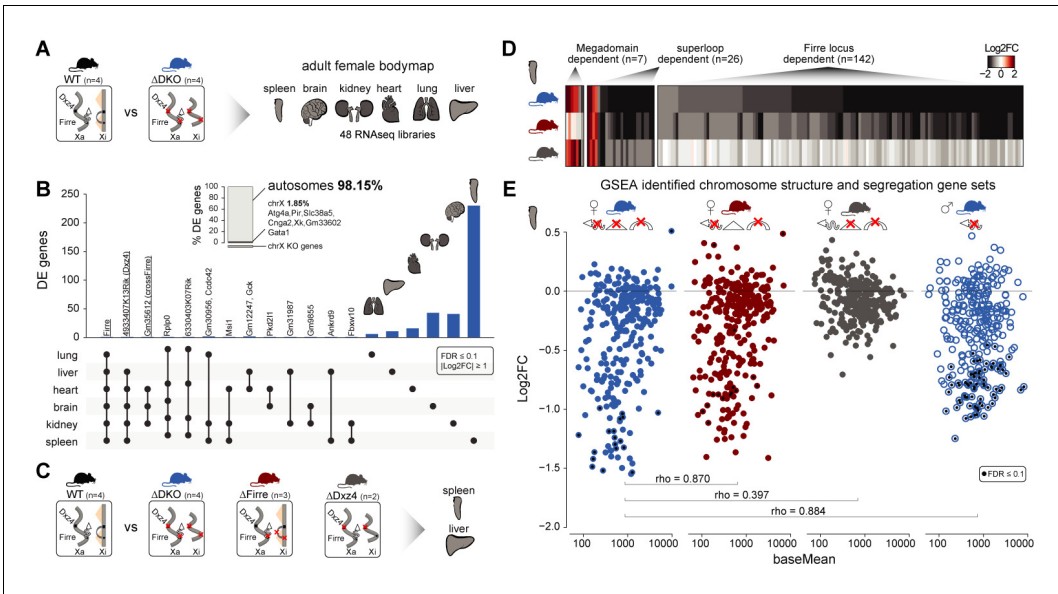

**Figure 4.** Homozygous deletion of *Firre* and *Dxz4* loci results in organ-specific expression changes on autosomes. (**A**) Cartoon illustrating the structural differences of the X chromosome between wildtype and DKO female mice (left), and the organs collected from 4 WT and 4 DKO six-weeks old adult females to generate the transcriptomic bodymap (right). (**B**) Overview of the differentially expressed genes in the female bodymap and their overlap across the six organs (DEseq2: FDR ≤ 0.1, |log2FC| ≥ 1). The bar plot shows the proportion of differentially expressed genes between autosomes and the X chromosome. Genes that are directly affected by the deletion are underlined (*Firre*, *Dxz4* transcript (*4933407K13Rik*) and *crossFirre* (*Gm35612*, antisense to *Firre*). (**C**) Cartoon illustrating the structural differences of the X chromosome between WT and each of the KO strains (left), and the SKO organs collected from six-weeks old adult females (Firre = 3, Dxz4 = 2) (right). (**D**) Heatmap showing the fold changes of megadomain, superloop and *Firre* locus dependent gene sets in the spleen. (**E**) MA plot for all KO strains of the genes extracted from the top five gene sets (GSEA analysis) identified in the DKO and *Firre* SKO spleen (more details in the Materials and methods section and ***Supplementary file 1*** sheet I). The black dot indicates significant differentially expressed genes (DEseq2: FDR ≤ 0.1).
DOI: https://doi.org/10.7554/eLife.47214.010

The following figure supplements are available for figure 4:

**Figure supplement 1.** Analyzing adult female organs carrying a homozygous deletion of *Firre* and *Dxz4*.
DOI: https://doi.org/10.7554/eLife.47214.011

**Figure supplement 2.** Analyzing adult female organs carrying a homozygous deletion of *Firre* and/or *Dxz4*.
DOI: https://doi.org/10.7554/eLife.47214.012

genes across organs (***Figure 4B***, ***Supplementary file 1*** sheet F). The highest number of differentially expressed genes was detected in the spleen (n = 239, compared to the rest of the organs that on average had 30 dysregulated genes).

To validate and categorize the dysregulated genes identified in the DKO into superloop, megadomain, and *Firre* locus-dependent gene sets, we sequenced the spleen and the liver from independently generated *Firre* and *Dxz4* SKO animals (***Figure 4C***). We hypothesized that dysregulated gene sets that are: (1) shared across the three strains are superloop specific (2) shared between *Dxz4* SKO and DKO are megadomain-dependent and (3) shared between *Firre* SKO and the DKO are *Firre* locus specific. To identify gene sets for these categories, we selected the genes dysregulated in at least one of the three KO strains where the direction of the expression change in either one of the single KO agrees with that of the DKO (|log2FC| ≥ 1). We first used this approach for the liver, an organ with a low number of differentially expressed genes in the DKO, and identified 15 megadomain, four superloop, and 4 *Firre* locus dependent genes (***Figure 4—figure supplement 2A***, ***Supplementary file 1*** sheet G). Notably, without applying the fold change cutoff, we observe

megadomain dependence of *Xist*, which is significantly upregulated in the DKO brain and kidney, as well as in the *Dxz4* SKO spleen (mean upregulation 30%, *Figure 4—figure supplement 2B*).

We next applied the same approach to the spleen, the organ with the highest number of differentially expressed genes in the DKO, and identified seven megadomain (4%), 26 superloop (14.8%) and 142 *Firre* locus (81.1%) dependent genes (*Figure 4D*, *Supplementary file 1* sheet H). The *Firre* locus dependent gene set—the largest group—contains only downregulated genes, including the gene *Ypel4* that was previously described to form interchromosomal interactions with the *Firre* locus (*Hacisuleyman et al., 2014*; *Maass et al., 2018*).

By gene set enrichment analysis (GSEA), we find that the top *Firre* SKO and DKO enriched gene sets are almost identical, sharing downregulated gene sets involved in chromosome structure and segregation (*Figure 4—figure supplement 2C*, *Supplementary file 1* sheet I). Genes extracted from the enriched gene sets identified in the *Firre* SKO and DKO share a similar pattern of downregulation, which was not observed in the *Dxz4* SKO (*Figure 4E*). In addition, fold changes in *Firre* SKO and DKO are strongly correlated (spearman rho = 0.87, p-value$<2.2*10^{-16}$), indicating that the *Firre* locus is the main driver for downregulation of these gene sets (*Figure 4E*). To test whether this molecular phenotype can be uncoupled form X inactivation, we performed the same analysis in DKO male spleens. We found a similar pattern of downregulation as observed in females (*Figure 4E*). Across all strains carrying the *Firre* locus deletion, the enriched gene sets share the majority of genes with the greatest degree of dysregulation (*Supplementary file 1* sheet I). Taken together these findings point to the *Firre* locus, independently of X inactivation, as the main driver of these autosomal expression signatures.

## Discussion

The *Firre* and *Dxz4* loci provide the platform for the formation of the X chromosome mega-structures and have been extensively studied in cell lines modeling random XCI (*Rao et al., 2014*; *Horakova et al., 2012*; *Deng et al., 2015*; *Giorgetti et al., 2016*; *Bonora et al., 2018*; *Froberg et al., 2018*; *Darrow et al., 2016*; *Barutcu et al., 2018*). Here we addressed the in vivo role of these elements by generating mice carrying a single or double deletion of these loci. In agreement with previous in vitro studies, we find that the loss of these loci in vivo does not affect random XCI. The lack of dysregulated X-linked genes in adult organs suggests that these mega-structures may also not be important for long-term maintenance of the Xi chromosome. Moreover, by studying the placenta and the visceral yolk sac we were able to show for the first time that loss of these loci also does not affect imprinted XCI.

Remarkably, deletion of these loci results in reproducible organ-specific expression changes on autosomes suggesting that structural changes of the Barr body may lead to autosomal gene dysregulation. Indeed, crosstalk between autosomes and the X chromosome has been proposed as a mechanism for X inactivation counting (*Rastan, 1983*). Whether these changes are directly regulated by the Barr body structure remains to be investigated. The largest transcriptional effect on autosomes is *Firre* locus-dependent and X inactivation-independent, suggesting a role for the *Firre* RNA or DNA locus in autosomal gene regulation. The major function of these dysregulated genes appears to be in ontologies associated with chromosome structure and segregation, which is in line with the known role for *Firre* lncRNA in nuclear organization (*Hacisuleyman et al., 2014*). This relationship points to an RNA dependent role, which may be either direct or indirect, on autosomal gene regulation. Collectively, our results indicate that the X-linked loci *Firre* and *Dxz4*, are involved in autosomal gene regulation rather than XCI biology in vivo.

### Accession codes

Sequence data and alignments have been submitted to the Gene Expression Omnibus (GEO) database under accession code GSE127554.

## Materials and methods

### Mouse strains

Mice were housed under controlled pathogen-free conditions (Harvard University's Biological Research Infrastructure). C57BL/6J (Bl6), B6D2F1/J (F1 Bl6 and DBA) and CAST/Ei (CAST) mice were purchased from the Jackson Laboratory.

The *Firre* deletion mouse (deletion mm10: chrX:50,555,286–50,637,116) was generated by inserting LoxP sites flanking the *Firre* gene body into JM8A ESCs (Bl6 background) (*Hacisuleyman et al., 2014*). Upon injecting the ESCs into blastocysts, the resulting founder mouse was crossed with CMV-CRE (B6.C-Tg(CMV-cre)1Cgn/J, JAX Stock no: 006054. The strain was generated from BALB/c-I ESCs and backcrossed to the C57BL/6J background for 10 generations). The resulting *Firre* SKO mouse was subsequently backcrossed three times with Bl6, as described in detail in *Lewandowski et al. (2019)* (*Figure 1—figure supplement 1a*).

The *Dxz4* single deletion strain (chrX:75,721,164–75,764,733 mm10) was generated by co-injecting Cas9 mRNA (200 ng/µl) together with two guide RNA's that span the *Dxz4* locus (50 ng/µl each) into pronuclear stage 3 (PN3) zygotes isolated after mating super ovulated B6D2F1 female mice (Jackson labs) with Bl6 males as previously described (*Wang et al., 2013*) (*Figure 1—figure supplement 1a*).

The *Firre-Dxz4* double deletion strain (*Dxz4* deletion chrX:75,720,836–75,764,839, *Firre* deletion same as for *Firre* SKO), was generated by piezo-assisted Intracytoplasmic sperm injection of *Firre* SKO sperm into B6D2F1 oocytes (protocol described in *Yoshida and Perry, 2007*). Cas9 mRNA was co-injected with the two *Dxz4* locus spanning gRNAs as described above (*Figure 1—figure supplement 1a*).

Embryos were cultured to the blastocyst stage, transferred into pseudopregnant CD-1 strain females (Charles River), and brought to term. Protospacer sequences (shown in *Figure 1—figure supplement 1a*) were identified using ChopChop (*Labun et al., 2016*) and single guide RNAs were synthesized from T7 promoter containing oligonucleotides using the MEGAshortscript in vitro transcription system (Invitrogen). Founder mice of the *Dxz4* SKO and DKO strains were backcrossed at least two times with C57BL/6J to remove strain background or CRISPR-Cas9 off-target effects.

In order to control for strain background, all wildtype control mice were obtained from backcrossing of founder mice with C57BL/6J to match the KO strain background.

### Tissue isolation and library preparation

To determine whether the deletion of *Firre*, *Dxz4* or *Firre-Dxz4* impact random or imprinted X chromosome inactivation, we collected embryonic day 12.5 brains, placentas and visceral yolk sac from reciprocal F1 crosses between the deletion strains and CAST/EiJ. For the brain and the placenta, we collected samples for all three strains from the forward cross (3 males and three females for WT and maternal deletion) and reverse cross (three females for WT and paternal deletion). For the placenta DKO we added an additional replicate of the maternal and paternal deletion. In addition, we collected visceral yolk sac samples from the DKO forward cross (three females WT and maternal deletion) and reverse cross (three females WT and paternal deletion). To test whether DNA methylation levels are altered in the absence of *Firre* and *Dxz4 on* either Xa *or* Xi, we collected placentas from the DKO forward cross (one female WT and two maternal deletion) and reverse cross (two females WT and two paternal deletion) to perform reduced representation bisulfite sequencing (RRBS) sequencing. To reduce the amount of maternal contamination we removed the decidua of the placentas. For the *Firre-Dxz4* adult bodymap, we collected the spleen, brain, kidney, heart, lung and liver from 6 weeks old female mice carrying a homozygous double deletion (4 WT and 4 DKO replicates). To validate and classify the dysregulated genes form the DKO, we collected liver and spleen from independent generated female SKO strains (*Firre* 3 SKO and *Dxz4* 2 SKO replicates). To test whether the molecular phenotype observed in the female spleen can be uncoupled form X inactivation, we collected 6 weeks old spleens from DKO males (2 WT and 2 DKO replicates).

The collected tissues were snap frozen and stored at −80°C until further process. RNA was extracted from TRIzol lysates using RNeasy mini columns (Qiagen). The Illumina TruSeq kit was used to create polyA$^+$ libraries from total RNA. We generated strand-specific libraries for the F1 placentas and brains (TruSeq stranded Illumina) and non-strand-specific TruSeq libraries for the adult organs

and the visceral yolk sac. Libraries were quantified using a Qubit 2.0 Fluorometer, run on an Agilent 2100 Bioanalyzer to assess purity and fragment size, and sequenced on a HiSeq 2500 at Harvard University's Bauer Sequencing Core (75 bp paired end).

Genomic DNA for reduced representation bisulfite sequencing (RRBS) was quantified using a Qubit 2.0 Fluorometer, and quality-assessed on an Agilent 2200 TapeStation D1000 ScreenTape. RRBS was performed on 10 ng of each sample using the NuGen Ovation RRBS Methyl-Seq System following the manufacturer's recommendations except that barcoded adapter-ligated samples were pooled in groups of 8 immediately prior to Bisulfite Conversion with the Qiagen EpiTect Fast Bisulfite Conversion kit. Library pools were purified with a 1X Agencourt RNA XP bead clean-up and sequenced on a HiSeq 4000. Sequenced reads were aligned to the mm10 reference genome using BSmap (*Xi and Li, 2009*) and methylation states were extracted using the MOABS mcall module (*Sun et al., 2014*).

## RNA-seq alignment and analysis

The RNA data were aligned with STAR by using specific parameters to exclude reads mapping to multiple locations (STAR version 2.5.0 c: –outFilterMultimapNmax 1) (*Dobin et al., 2013*). The read counts for every isoform within the RefSeq gene annotation (downloaded February 2018) were calculated by using the Python script htseq-count (HTSeq version 0.6.1) (*Anders et al., 2015*).

Differential expression analysis was performed with the assumption of negative binomial distribution of the read counts and empirical estimation of variance by using the R packages DESeq2 (version 1.22.1) (*Love et al., 2014*) and fdrtool (*Strimmer, 2008*). Genes were called significant if their FDR-adjusted p-values were smaller or equal than 0.1.

Allele-specific expression was detected from RNA-seq by using the Allelome.PRO, as described in detail in *Andergassen et al. (2015)*. Allelome.PRO uses the information of characterized single-nucleotide polymorphisms (SNPs) to assign sequencing reads to the corresponding strain in F1 crosses. For the SNP annotation we first extracted 20,606,390 high confidence SNPs between the CAST/EiJ (CAST) and C57BL6NJ (Bl6) form the Sanger database as described previously (*Andergassen et al., 2015*; *Keane et al., 2011*). Although we backcrossed all the founder mice to Bl6, to exclude potential strain background confounding effects, we performed the allele-specific analysis using only CAST/Bl6 SNPs where the Bl6 allele was shared between DBA, BALB/C and 129 (Final SNP number: 15,438,314 SNPs). For the Allelome.PRO analysis we only included SNPs that are covered by at least two reads by setting the 'minread' parameter to 2.

## Superloop, megadomain, and *Firre* locus-dependent category assignment

To validate and categorize the dysregulated genes identified in the DKO bodymap analysis into superloop, megadomain, and *Firre* locus dependent gene sets, we selected genes dysregulated in at least one of the three KO strains (FDR $\leq$ 0.1) where the direction of the expression change in either one of the single KO agrees with that of the DKO (|log2FC| $\geq$ 1). Dysregulated gene sets that are: (1) shared across the three strains were categorized as superloop specific (2) shared between *Dxz4* SKO and DKO were categorized as megadomain-dependent and (3) shared between *Firre* SKO and the DKO were categorized as *Firre* locus specific (*Supplementary file 1* sheet G-H).

## GSEA analysis

The GSEA analysis was performed on DEseq2 test statistics with all GO gene sets (c5.all.v6.2.symbols) available from MSigDB (*Subramanian et al., 2005*) after mapping genes to gene sets by gene symbols. The calculation was performed in R using the CAMERA package (*Wu and Smyth, 2012*). A gene set was called significant if the FDR-adjusted p-value (Benjamini and Hochberg method) is less than or equal to 0.1. Top five enriched dysregulated gene sets identified in the DKO and *Firre* SKO spleen (*Figure 4—figure supplement 2C*), were extracted (*Supplementary file 1* sheet I) for each of the KO strains.

## Acknowledgements
We thank Philipp Maass, Marta Melé, Rasim Barutcu, Kaia Mattioli, Gabrijela Dumbovic and Quanah Hudson for stimulating discussions and critical reading of the manuscript. Nydia Chang for assistance in the mouse facility. Sequencing was performed at the Bauer Core Facility at Harvard University.

## Additional information

### Funding

| Funder | Grant reference number | Author |
|---|---|---|
| National Institutes of Health | P01 GM099117 | John L Rinn<br>Alexander Meissner |
| Howard Hughes Medical Institute | Faculty Scholar | John L Rinn |
| Max-Planck-Gesellschaft | | Alexander Meissner |

The funders had no role in study design, data collection and interpretation, or the decision to submit the work for publication.

### Author contributions
Daniel Andergassen, Conceptualization, Resources, Data curation, Software, Formal analysis, Validation, Investigation, Visualization, Methodology, Writing—original draft, Writing—review and editing; Zachary D Smith, Conceptualization, Resources, Investigation, Methodology, Writing—review and editing; Jordan P Lewandowski, Resources, Methodology, Writing—review and editing; Chiara Gerhardinger, Conceptualization, Supervision, Investigation, Methodology, Writing—review and editing; Alexander Meissner, Conceptualization, Supervision, Funding acquisition, Writing—review and editing; John L Rinn, Conceptualization, Supervision, Funding acquisition, Project administration, Writing—review and editing

### Author ORCIDs
Daniel Andergassen (iD) https://orcid.org/0000-0003-1196-4289
Alexander Meissner (iD) https://orcid.org/0000-0001-8646-7469
John L Rinn (iD) https://orcid.org/0000-0002-7231-7539

### Ethics
Animal experimentation: Mice used in these studies were handled according to approved institutional animal care and use committee (IACUC) protocols (#28-21) of Harvard University. Procedures were performed in accordance with the National Institutes of Health guidelines for the Care and Use of Laboratory Animals.

### Decision letter and Author response
Decision letter https://doi.org/10.7554/eLife.47214.018
Author response https://doi.org/10.7554/eLife.47214.019

## Additional files

### Supplementary files
• Supplementary file 1. RNA and reduced representation bisulfite sequencing analysis. (**A**) Log2FC (lfcMLE) and adjusted p-values (padj) from DEseq2 differential expression analysis, computed by comparing female WT placentas of the forward cross (n = 8) with each of the deletion strains (deletion on Xa, n = 3–4) and female WT placentas of the reverse cross (n = 9) with each of the deletion strains (deletion on Xi, n = 3–4). (**B**) Log2FC (lfcMLE) and adjusted p-values (padj) from DEseq2 differential expression analysis, computed by comparing male WT placentas of the forward cross (n = 9) with each of the KO strains (n = 3). (**C**) Methylation levels as measured by reduced

representation bisulfite sequencing (RRBS) for WT placenta and DKO deletion on Xa or Xi. (**D**) Imprinted ratios of X-linked placenta genes for WT and each of the deletion strains (deletions on Xa or Xi, 0.5 = 100% maternal, −0.5 = 100% paternal). The allelic ratios for each replicate per genotype was combined by using the median. (**E**) Imprinted ratios of X-linked visceral yolk sac genes for WT and DKO (deletions on Xa or Xi, 0.5 = 100% maternal, −0.5 = 100% paternal). The allelic ratios for each replicate per genotype was combined by using the median. (**F**) Differentially expressed genes (DEseq2 differential expression analysis) in all tissues (bodymap) of DKO female and liver and spleen of SKO female. (**G-H**) Log2FC of megadomain, superloop and *Firre* locus dependent gene sets in the liver and spleen. (**I**) Top enriched gene sets identified in the DKO and *Firre* SKO spleen. (**J**) Information of every analyzed sample in this study.
DOI: https://doi.org/10.7554/eLife.47214.013

- Transparent reporting form DOI: https://doi.org/10.7554/eLife.47214.014

## Data availability

Sequence data and alignments have been submitted to the Gene Expression Omnibus (GEO) database under accession code GSE127554.

The following dataset was generated:

| Author(s) | Year | Dataset title | Dataset URL | Database and Identifier |
|---|---|---|---|---|
| Andergassen D, Meissner A, Rinn JL | 2019 | In vivo Firre and Dxz4 deletion elucidates roles for autosomal gene regulation | https://www.ncbi.nlm.nih.gov/geo/query/acc.cgi?acc=GSE127554 | NCBI Gene Expression Omnibus, GSE127554 |

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
