## [Decision Letter]

Thank you for submitting your article "in vivo Firre and Dxz4 deletion elucidates roles for autosomal gene regulation" for consideration by *eLife*. Your article has been reviewed by three peer reviewers, including Jeannie T Lee as the Reviewing Editor and Reviewer #1, and the evaluation has been overseen by Michael Eisen as the Senior Editor.

The reviewers have discussed the reviews with one another and the Reviewing Editor has drafted this decision to help you prepare a revised submission.

Summary:

The study generates single or double deletions of Firre and Dxz4 in mice, and show that despite the repeats being conserved in mammals, these mutants are viable, fertile and show no defect in random or imprinted XCI. Instead, the lack of Firre and Dxz4 results in dysregulated genes on autosomes in an organ-specific manner. Although the authors present mostly negative data, the manuscript would be of interest to the field and is significant because (1) it analyzes the phenotype in vivo in a mouse model, contrasting with previous papers that performed experiments only in cell culture. (2) The current manuscript also sheds light on a previous claim by Giorgetti et al., (2016) that Dxz4 is required for genes to escape from XCI. (3) It is revealed that the loci also do not play a role in imprinted XCI. All three reviewers believe that the scope of the work is in principle in line with *eLife*. However, for publication in *eLife*, we would require several major revisions.

1) Better controls to rule out strain background or off-target artifacts, when looking at changes in autosomal gene expression. (See reviewer 3 notes).

2) More backcrosses and RNA-seq of placenta, as it is the only tissue with non-random XCI, and therefore the only tissue where the authors can distinguish between effects stemming from Firre/Dxz4's roles on the inactive vs. the active X. (See reviewer 2).

3) Direct versus indirect effects of Dxz4/Firre on autosomal expression. If the authors would like to claim a direct effect, CHART analyses would be required. Otherwise, the statements should be toned down. (See reviewer 1 notes).

4) The 2 main figures should be broken up. (See reviewer 1 notes).

5) Revision of statistical analyses. (See reviewer 2 notes).

6) Finally, to the extent possible, please try to address the additional concerns of reviewers 2 and 3.

*Reviewer #1:*

The paper from Andergassen et al., is a succinct report made up of mostly negative findings and confirms the negative results of several past publications. Although the knockouts rule out a role in XCI, the report is significant in that (1) it analyzes the phenotype in vivo in a mouse model, contrasting with previous papers that performed experiments only in cell culture; In the majority of past reports, it was shown that the conserved X chromosome "mega-structures" controlled by the Firre and Dxz4 alleles are not required for XCI in cell lines. (2) The current manuscript also sheds light on a previous claim by Giorgetti et al., (2016) that Dxz4 is required for genes to escape from XCI. No effect on gene escape is seen by Andergassen et al., in vivo. (3) The loci also do not play a role in imprinted XCI.

The authors generate mice carrying a single or double deletion of Firre and Dxz4, and show that despite the repeats being conserved in mammals, these mutants are viable, fertile and show no defect in random or imprinted XCI. One positive finding is that the lack of Firre and Dxz4 results in dysregulated genes on autosomes in an organ-specific manner. By comparing the dysregulated genes between the single and double deletion, they categorize superloop, megadomain, and Firre locus-dependent gene sets and see that Firre deletion has the greatest effect on autosomal expression signatures. The manuscript is within the scope of papers published in *eLife* and would be of interest to the field. Overall, the analysese are done well and the conclusions are mostly supported by the data. However, I recommend several revisions:

1) While I accept that there are changes in autosomal gene expression, I am less convinced that the effects are direct. I am also not convinced that X-chromosomal superstructures directly affect autosomal gene expression. If the authors would like to claim a direct role, they must perform additional analyses, including – for example – CHART analysis to correlate changes at specific loci with binding of the RNA, or 3C or Hi-C to determine if the loop domains interact with autosomal gene targets. Do these autosomal targets include the loci that the authors previously showed to interact with Firre?

2) If the authors do not wish to include additional work, they should tone down the conclusions regarding autosomal changes and acknowledge the possibility of indirect effects, per point 1.

3) The figures are very dense, difficult to read (panels are small), and should be broken up into additional figures. *eLife* allows more than 2 display panels. At least one figure should be devoted to a more careful and deeper analysis of XCI gene escape, since the lack of effect on escape is a major point of the paper. And another figure should likewise be devoted to a deeper analysis of imprinted XCI, since this is another major point of the paper.

*Reviewer #2:*

The manuscript by Andergassen et al., describes the generation of mouse single and double knockouts of two X-linked macrosatellites, Firre and Dxz4. The authors analyze sex ratio, litter size and expression phenotypes of these three knockouts relative to wild-type, i.e. F1 hybrids of musculus x castaneous crosses. Whether and how these macrosatellites may participate in X chromosome inactivation or escape from XCI is an important question, due to previously described roles of Firre (not referenced PMID: 26048247 and PMID:25887447) and Dxz4 (not referenced PMID:26248554) in gene expression and inactive X chromosome conformation, respectively. However, as presented, in vivo defects were non-comprehensively analyzed, and gene expression experiments improperly controlled and somewhat over-interpreted:

1) The authors main conclusion, that there is no sex ratio distortion in the double ko, is partially undermined by a miscalculated p-value (should be 0.2712 under a p=0.5 cumulative binomial) and only six reported litters. This table should also include the outcomes of the wildtype control and single homozygous ko matings.

2) Differences in the genetic background of wild-type (not specified), single knockouts (after only two backcrosses: ~12.5% BALB/C genome for Firre, and ~6.25% DBA for Dxz4), and the double ko (variable ~1.6% BALB/C and ~6.25% DBA), are not taken into account in the gene expression analysis. This would be a non-issue with more backcrosses to C57BL/6J, at least five (see Silver, mouse genetics book).

3) Low and inconsistent numbers of replicates in the RNA-seq that reduce statistical power. This is especially problematic for attributing differential expression to loss of megadomains and superloop, because this analysis depended on comparing across multiple differential gene lists, which shrink with low statistical power (see only two Dxz4 ko replicates). This could be addressed by additional replicates from the single ko's, to get at least four across the board for liver, spleen and placenta.

4) Homozygous animals also lack Firre RNA (and possibly Dxz4-associated transcripts) expressed from the active X. The analysis here assumes that there are no confounding interactions between differential genes attributed to one group or another. A deeper analysis of the placental data in heterozygous F1 hybrid animals, where such interactions can be excluded, would be preferable and likely yield more insight towards identifying megadomain- or superloop-specific gene expression, if any.

In conclusion, while the authors present a worthwhile question and set of experiments, more careful analysis and interpretation is necessary to address functions of Firre and Dxz4 in XCI and development. At minimum, the placental RNA-seq should be repeated and fully analyzed after more than 5 backcrosses, with at least 4 replicates for each of the 4 genotypes presented. Additional novelty could come from three interesting observations reported here: (1) enrichment for differential genes implicated in faithful chromosome segregation in mice lacking Firre expression, (2) Xist upregulation by 30% in Dxz4 single and double ko's (Figure 4—figure supplement 2B), and (3) shift in allelic ratio in the spleen of Dxz4 ko's (Figure 3—figure supplement 2B). Are there specific genes driving this or is the deletion skewing XCI?

*Reviewer #3:*

In their study, Andergassen et al., describe mouse mutants for the noncoding RNA-gene Firre and the Dxz4 Megadomain boundary element, which are important for X-chromosome 3D-structure. It has been an important question in the X-inactivation field, what the functional significance of these elements is and what their impact is on X-chromosome inactivation. Recent studies using cell lines have already demonstrated that deletion of these elements does not affect X-chromosome inactivation (XCI) in vitro. However, the current study is the first one to assess their importance in vivo and during imprinted XCI in the placenta. The data regarding a lack of an XCI phenotype in Firre and Dxz4-deletion mice is therefore important for the X-inactivation field.

The second main message of the paper is that nevertheless changes in autosomal gene expression are observed. Only few changes appear to be dependent on Dxz4-deletion, however more are observed in the Firre KO, which is expected from its role as a non-coding RNA, which could also have function away from the X-chromosome. However, as mentioned in my specific comments below, off-target or strain background effects can at the moment not be completely ruled out. Therefore, the authors should address these points, to ensure that the autosomal changes in gene expression are indeed caused by the Firre and Dxz4 mutation and are not artifacts.

Essential revisions:

1) The authors should perform Southern blotting and/or other assays to confirm that no autosomal off-target mutations were introduced when targeting the Firre and Dxz4 loci. PCR-screening alone as done at the moment (Figure 1—figure supplement 1) is not sufficient to rule that out especially considering the low number of back-crossings and the variability between replicates when looking at changes in autosomal gene expression (Figure 2D). This would be important especially as a main message of the paper is that there are Firre/Dxz4-dependent changes in autosomal gene expression.

2) Figure 2: The different knockout alleles have been generated in mixed strain backgrounds (129, BALB/C, DBA) and only relatively few backcrosses seem to have been performed to pure B6 background (>2). Where did the wildtype mice come from, to which autosomal gene expression levels were compared in the adult expression bodymap? Were they of equivalent mixed strain background, or were wildtype controls of pure strain background used? It is important that controls of matched strain background were used throughout the study, in order to avoid differences in autosomal gene expression due to strain-background effects.

3) Figure 3—figure supplement 2B and Figure 4—figure supplement 4C: Xist RNA seems to be upregulated dependent on Megadomain deletion (Figure 4—figure supplement 4C) and the allelic-ratio seems to skew random X-inactivation in the brain more towards the M Musculus (Dxz4-deleted) allele in Dxz4 and Firre/Dxz4 KO mice. The authors should check, if Xist skewing and allelic ratios of X-linked genes are significantly changed during random XCI in the Dxz4 and Firre/Dxz4 KOs.

4) Figure 2D: The megadomain-specific upregulation of urinary cluster genes in liver seems to be very variable between replicates. It would be good to add a third replicate for delta Dxz4 as at the moment one replicate shows strong upregulation, while the second one doesn't. Also, in the Firre/Dxz4 double-mutants half of the replicates show an effect while others don't. How do the authors explain this discrepancy between replicates?

[Editors' note: further revisions were requested prior to acceptance, as described below.]

Thank you for resubmitting your work entitled *"*in vivo *Firre* and *Dxz4* deletion elucidates roles for autosomal gene regulation" for further consideration at *eLife*. Your revised article has been favorably evaluated by Michael Eisen (Senior Editor), a Reviewing Editor, and three reviewers.

Reviewer 1 is satisfied with the revisions (no comments attached). Reviewer 3 also feels that the paper is much improved but remains concerned about strain background. We know that it may be difficult to completely rule out strain background issues, and thus do not consider this to be an impediment to publication. However, reviewer 2 has a number of remaining concerns. In particular:

1) Please include a supplementary table listing all RNA-seq samples as reviewer 2 suggested.

2) Please present a Venn diagram listing the CAST/BL6 -common and specific escaping genes in placental RNA-seq.

3) Please add a p-value for that difference in median X-linked expression and Xist differences.

4) Please report Dxz4 SKO and DKO sex ratios in Figure 1B.

5) Reviewer 2 also requested that you please re-analyze genome-wide differential expression without log2 FC cutoff in the placental RNA-seq data to distinguish the Firre locus from superloop and megadomain differentially expressed genes (if any) for Figure 4.

6) Resolve the incongruence between the genes listed as commonly differentially expressed in Figure 4 and the gene lists in the supplementary file. Please see comments below for more details.

*Reviewer #2:*

The revised manuscript by Andergassen et al., adds new placental RNA-seq data that was requested by reviewers so that the authors could exploit imprinted XCI to untangle the effects of Firre and Dxz4 deletions residing on the active vs. the inactive X. This is impossible to do in tissues with random XCI as analyzed in the "bodymap". The authors should state clearly that the conclusions of their bodymap data are possibly impacted by random XCI and unaccounted genetic variation (raised in the prior round of review). There appear to be very few consistently diff. expressed genes for any of the deletions in the placenta or other tissues (except the spleen). Moreover, the authors conclude that XCI and escape in the placenta are not affected, but apply a minimal log2FC cutoff of 1, the maximal differential that could be expected if XCI or escape are impacted (going from 1 to 2 active copies or vice versa). Allelic analyses which address this question directly is only presented in aggregated figures, supplementary tables, or overly compressed log-scales, obscuring gene-specific changes. These issues need to be addressed, not by new experiments, but a more in-depth and transparent gene expression analysis:

1) Please include a supplementary table listing all RNA-seq samples (indicating replicate groups) with read depths and% reads aligned. Please include PCA plots of all samples in this analysis, next to or in place of supplemental heatmaps. Supplementary tables on placental RNA-seq showed some genes with high log2FC that failed the significance cutoff – is this a replicate/variance issue? Was batch-correction applied to minimize the impact of different strain backgrounds? There is insufficient information in the methods to replicate this analysis.

2) Seeing more escaping genes on the CAST inactive X is an interesting observation, but not presented in meaningful depth. Allele-specific analysis of genes subject to and escaping XCI in the placental RNA-seq should be plotted on log2FC scale from -3 to +3 with Firre as an outlier (Figure 3A, rather than from -12 to +12). And in Figure 3B and C, there should be a Venn diagram listing the CAST/BL6 -common and specific escaping genes in placental RNA-seq.

3) Both DKO and Dxz4 show increased Xist expression and DKO animals have a very significant change towards more BL6-biased random XCI (Figure 3—figure supplement 2, please add a p-value for that difference in median X-linked expression). Figure 4—figure supplement 2B should be moved to main Figure 3, as Xist upregulation in the absence of Dxz4 appears to be consistent across the animal. Non-coding genes regulating Xist in cis and skewing XCI were not listed in the supplement – were any of them changed? Please include these genes the supplementary tables (use Gencode reference rather than RefSeq if necessary). In view of Xist and XCI skewing, it is essential that Dxz4 SKO and DKO sex ratios are reported in Figure 1B.

4) Please re-analyze genome-wide differential expression without log2FC cutoff in the placental RNA-seq data to distinguish Firre locus from superloop and megadomain differentially expressed genes (if any). Include this info in Figure 4, and discuss in the context of the other random XCI bodymap & GSEA analysis. If Firre is only relevant in the spleen, please state so clearly.

5) There is incongruence between the genes listed as commonly differentially expressed in Figure 4 and the gene lists in the supplementary file (only 8 genes match the table & the Venn diagram in 4E). And only 45 genes are diff. expressed in the spleens of both Firre and DKO mice. Please indicate clearly which GSEA-enriched gene sets originated from which comparisons (update Venn in Figure 4E), and include info listing these enriched diff. genes in the updated supplementary tables. There is insufficient information here to replicate this analysis.

*Reviewer #3:*

In their revised version the authors have added some new data (visceral yolk sac data, DNA-methylation data, more replicates) and addressed many of my concerns in changing the wording and splitting the figures, making the manuscript more readable. They have not completely addressed the concerns related to the mixed strain background, which is a main weakness of the paper. Nevertheless, the authors have explained the situation in more detail, providing indirect (but no new experimental) arguments that the phenotypes observed are indeed due to the gene modifications and not due to strain background effects. Some additional doubts remain about the genotyping strategies used (PCR only – cannot rule out off-target mutations elsewhere in the genome).

Nevertheless, the paper has generally improved from the initial version and adds an interesting piece of information to the X-inactivation field regarding functional non-requirement of Firre and X-megadomains for X-inactivation in vivo, but having rather potential functions in direct or indirect autosomal gene regulation.

---

## [Author Response]

Summary:The study generates single or double deletions of Firre and Dxz4 in mice, and show that despite the repeats being conserved in mammals, these mutants are viable, fertile and show no defect in random or imprinted XCI. Instead, the lack of Firre and Dxz4 results in dysregulated genes on autosomes in an organ-specific manner. Although the authors present mostly negative data, the manuscript would be of interest to the field and is significant because (1) it analyzes the phenotype in vivo in a mouse model, contrasting with previous papers that performed experiments only in cell culture. (2) The current manuscript also sheds light on a previous claim by Giorgetti et al. (2016) that Dxz4 is required for genes to escape from XCI. (3) It is revealed that the loci also do not play a role in imprinted XCI. All three reviewers believe that the scope of the work is in principle in line with eLife. However, for publication in eLife, we would require several major revisions.1) Better controls to rule out strain background or off-target artifacts, when looking at changes in autosomal gene expression. (See reviewer 3 notes).

We agree with the reviewer comments that the relatively low number of backcrossing (fewer than 5) of the founder mice with the C57BL/6J strain could result in strain background differences between wild type and KO strains. To control for potential strain background or off-target effects artifacts, we performed all necessary control experiments in our initial submission which we will highlight and discuss in more detail below.

a) All the wildtype controls samples that we used for RNA-seq originate from backcrossing of the *Firre, Dxz4* and double deletion founder mice with the C57BL/6J strain in order to ensure a matching genetic background to the KO strains and, thus, to allow for proper comparison between wildtype and KO strains. To clarify this point, we added the following sentence to the “Materials and methods” section (Mouse strains): “In order to control for strain background, all wildtype control mice were obtained from backcrossing of founder mice with C57BL/6J to match the KO strain background.”

b) To minimize potential strain background or off-target effects, for the *Firre-Dxz4* double KO bodymap we used four biological replicates per group, collected from different litters.

c) The dysregulation of autosomal genes that was observed in the DKO mice, was confirmed in independently generated SKO mice, strongly arguing for a specific effect of the deletions, rather than strain background or off-target effects. For example, the largest spleen-specific transcriptional effect on autosomes detected in the DKO bodymap was also detected in independently-generated *Firre* SKO but not in the *Dxz4* SKO. Thus, the observed autosomal genes dysregulation can be confidently attributed to the specific deletion of the *Firre* locus rather than to potential background strain differences or off-target effects.

2) More backcrosses and RNA-seq of placenta, as it is the only tissue with non-random XCI, and therefore the only tissue where the authors can distinguish between effects stemming from Firre/Dxz4's roles on the inactive vs. the active X. (See Reviewer 2)

In addition to the above examples that strongly suggest background is not influencing our conclusions, it is important to note that the suggested backcrosses would take approximately a year. At the very least this would require redoing the entire sample collection and sequencing again.

In the placenta we had not observe dysregulation of X-linked genes or gene escape in the presence of the deletions on either Xa or Xi. We have now included one additional placenta biological replicate of the DKO (deletion on Xa and Xi) and confirmed that the deletion of these loci does not have an effect on the X chromosome (new Figure 3). To further investigate whether the deletions have any impact on DNA methylation levels of the inactive X chromosome, we have now also performed reduced representation bisulfite sequencing (RRBS) on placentas lacking both *Firre* and *Dxz4* on Xa or Xi. We found similar methylation levels as in wildtype, providing further evidence that deletion of these loci does not affect X chromosome biology (new Figure 3—figure supplement 1).

The reviewer also commented that the placenta was the only tissue in our dataset undergoing non-random XCI and we agree that it is worth examining another tissue with non-random XCI. Thus, to further explore the role of non-random XCI in *Firre* and *Dxz4* mutants, we collected the visceral yolk sac, an extra-embryonic tissue that, similarly to the placenta, undergoes imprinted XCI. We performed RNA-seq analysis of the visceral yolk sac and as observed in the placenta, we did not detect dysregulation of X-linked genes or gene escape in the presence of the deletions either on Xa or Xi (new Figure 3) a further evidence that the lack of the *Firre* lncRNA or the mega-structures does not affect imprinted X chromosome inactivation.

3) Direct versus indirect effects of Dxz4/Firre on autosomal expression. If the authors would like to claim a direct effect, CHART analyses would be required. Otherwise, the statements should be toned down. (See reviewer 1 notes).

We agree with the reviewer that the effect could be either direct or indirect. Indeed, we had acknowledged that further studies are needed to establish whether the effect of the megastructures on autosomal genes regulation is direct or indirect. With regard to the autosomal targets of *Firre* RNA that we reported previously (*Slc25a12, Ypel4, Eef1a1, Atf4 and Ppp1r10* in male mouse embryonic stem cells, PMID: 24463464), we did find that *Ypel4* is downregulated in the spleen of both DKO and *Firre* SKO mice. We have now highlighted this finding in the revised manuscript. While this finding would suggest a direct role of *Firre* RNA in at least some of the expression changes, we agree that our experimental approach does not permit to discriminate between direct and indirect effects. To clarify this point, we have revised the text in the Discussion section as follow: “The major function of these dysregulated genes appears to be in ontologies associated with chromosome structure and segregation, which is in line with the known role for *Firre* lncRNA in nuclear organization. This relationship points to an RNA dependent role, which may be either direct or indirect, on autosomal gene regulation.”

4) The 2 main figures should be broken up. (See reviewer 1 notes).

To reduce complexity, we broke up the two main figures into 4 figures. As suggested by the reviewers, new Figure 3 is now dedicated to the analysis of imprinted XCI and gene escape, and includes the new results from the visceral yolk sac.

5) Revision of statistical analyses. (See reviewer 2 notes).

Thank you very much for the observation, we now reported the correct p-value in new Figure 1B. We also added the sex ratio, and provided additional details regarding how calculations were performed in the figure legend.

6) Finally, to the extent possible, please try to address the additional concerns of reviewers 2 and 3.

See point by point reviewer response.

Reviewer #1:The paper from Andergassen et al., is a succinct report made up of mostly negative findings and confirms the negative results of several past publications. Although the knockouts rule out a role in XCI, the report is significant in that (1) it analyzes the phenotype in vivo in a mouse model, contrasting with previous papers that performed experiments only in cell culture; In the majority of past reports, it was shown that the conserved X chromosome "mega-structures" controlled by the Firre and Dxz4 alleles are not required for XCI in cell lines. (2) The current manuscript also sheds light on a previous claim by Giorgetti et al., (2016) that Dxz4 is required for genes to escape from XCI. No effect on gene escape is seen by Andergassen et al., in vivo. (3) The loci also do not play a role in imprinted XCI.The authors generate mice carrying a single or double deletion of Firre and Dxz4, and show that despite the repeats being conserved in mammals, these mutants are viable, fertile and show no defect in random or imprinted XCI. One positive finding is that the lack of Firre and Dxz4 results in dysregulated genes on autosomes in an organ-specific manner. By comparing the dysregulated genes between the single and double deletion, they categorize superloop, megadomain, and Firre locus-dependent gene sets and see that Firre deletion has the greatest effect on autosomal expression signatures. The manuscript is within the scope of papers published in eLife and would be of interest to the field. Overall, the analysese are done well and the conclusions are mostly supported by the data. However, I recommend several revisions:1) While I accept that there are changes in autosomal gene expression, I am less convinced that the effects are direct. I am also not convinced that X-chromosomal superstructures directly affect autosomal gene expression. If the authors would like to claim a direct role, they must perform additional analyses, including – for example – CHART analysis to correlate changes at specific loci with binding of the RNA, or 3C or Hi-C to determine if the loop domains interact with autosomal gene targets. Do these autosomal targets include the loci that the authors previously showed to interact with Firre?2) If the authors do not wish to include additional work, they should tone down the conclusions regarding autosomal changes and acknowledge the possibility of indirect effects, per point 1.

We thank the reviewer for the comments. Indeed, we had acknowledged that the effect of the megastructures on autosomal genes regulation could be either direct or indirect (Discussion section of the first submission: “Whether these changes are directly regulated by the Barr body structure remains to be investigated”). With regard to the autosomal targets of *Firre* RNA that we reported previously (*Slc25a12, Ypel4, Eef1a1, Atf4 and Ppp1r10* in male mouse embryonic stem cells, PMID: 24463464), we found that *Ypel4* is downregulated in the spleen of both DKO and *Firre* SKO mice. We have now highlighted this finding in the Results section of the revised manuscript, as follow: (“The *Firre* locus dependent gene set—the largest group—contains only downregulated genes, including the gene *Ypel4* that was previously described toform interchromosomal interactions with the *Firre* locus.”). While this finding would suggest a direct role of *Firre* RNA in at least some of the expression changes, we agree that our experimental approach does not permit to discriminate between direct and indirect effects. We have now clarified this point as indicated in “Essential revision” point 3.

3) The figures are very dense, difficult to read (panels are small), and should be broken up into additional figures. eLife allows more than 2 display panels. At least one figure should be devoted to a more careful and deeper analysis of XCI gene escape, since the lack of effect on escape is a major point of the paper. And another figure should likewise be devoted to a deeper analysis of imprinted XCI, since this is another major point of the paper.

Please refer to “Essential revision” point 4.

Reviewer #2:The manuscript by Andergassen, et al., describes the generation of mouse single and double knockouts of two X-linked macrosatellites, Firre and Dxz4. The authors analyze sex ratio, litter size and expression phenotypes of these three knockouts relative to wild-type, i.e. F1 hybrids of musculus x castaneous crosses. Whether and how these macrosatellites may participate in X chromosome inactivation or escape from XCI is an important question, due to previously described roles of Firre (not referenced PMID: 26048247 and PMID:25887447) and Dxz4 (not referenced PMID:26248554) in gene expression and inactive X chromosome conformation, respectively. However, as presented, in vivo defects were non-comprehensively analyzed, and gene expression experiments improperly controlled and somewhat over-interpreted:1) The authors main conclusion, that there is no sex ratio distortion in the double ko, is partially undermined by a miscalculated p-value (should be 0.2712 under a p=0.5 cumulative binomial) and only six reported litters. This table should also include the outcomes of the wildtype control and single homozygous ko matings.

Thank you very much for catching this error. We have now corrected it as indicated in point 5 of “Essential revision”.

Given that the male-female ratio of the DKO progeny was not significantly altered, we reasoned that testing for sex ratio skewing within the WT control group was not required. With regard to the SKO strains, the sex ratio of the *Firre* SKO, and corresponding WT, was evaluated as part of another recent study from our group (Lewandowski et al., 2019), and it was found to be normal. Based on the finding in the DKO and *Firre* SKO strains, we reasoned that a sex ratio skewing in the *Dxz4* SKO strain would also be unlikely.

2) Differences in the genetic background of wild-type (not specified), single knockouts (after only two backcrosses: ~12.5% BALB/C genome for Firre, and ~6.25% DBA for Dxz4), and the double ko (variable ~1.6% BALB/C and ~6.25% DBA), are not taken into account in the gene expression analysis. This would be a non-issue with more backcrosses to C57BL/6J, at least five (see Silver, mouse genetics book).

We agree with the reviewer of the importance to control for potential strain background differences. We addressed this point in “Essential revision” point 1. In addition, we would like to clarify that the *Firre* SKO mouse was generated by inserting LoxP sites flanking the *Firre* gene body into JM8A ESCs (PMID: 19525957) which are of Bl6 background and not 129xBl6 as we indicated in the original submission. The ESCs were then injected into blastocysts to obtain chimeras that were then backcrossed to Bl6 to obtain the founder mouse. The founder mouse was then crossed with CMV-CRE mice B6.C-Tg(CMV-cre)1Cgn/J, JAX Stock no: 006054. While the CRE JAX strain was generated from BALB/c-I ESCs, stock 006054 was backcrossed to the Bl6 background for 10 generations. Thus, the background strain is Bl6 and not BALB/c as we indicated in the original submission. We apologize for the confusion. The resulting *Firre* SKO mouse was subsequently backcrossed three times with Bl6, more details in Lewandowski et al. bioXriv 2019. Based on this consideration the background of the *Firre* SKO is almost 100% Bl6 (Figure 1—figure supplement 1A). The *Dxz4* single deletion strainwas generated by co-injecting Cas9 mRNA together with two guide RNA’s that span the *Dxz4* locus into pronuclear stage 3 (PN3) zygotes isolated after mating super ovulated B6D2F1 (50% Bl6 50% DBA) female mice with Bl6 males resulting in a founder mouse with 75% Bl6 background. The Founder mouse was then backcrossed two times with Bl6, and can thus be considered as 93.75% Bl6 background (Figure 1—figure supplement 1a). The *Firre-Dxz4* double deletion strain, was generated by piezo-assisted Intracytoplasmic sperm injection of *Firre* SKO sperm (Bl6 background) into B6D2F1 (50% Bl6 50%DBA) oocytes (at PN3, Cas9 mRNA was co-injected with the two *Dxz4* locus spanning gRNAs.) The resulting DKO founder mouse was then backcrossed two times with Bl6, and can thus be considered as 93.75% Bl6 (Figure 1—figure supplement 1A). All the wildtype controls samples that we used for RNA-seq originate from backcrossing of the *Firre, Dxz4* and DKO founder mice with the Bl6 strain in order to ensure a matching genetic background to the KO strains and, thus, to allow for proper comparison between wildtype and KO strains.

We have now provided a more detailed description of the generation of all strains, as well as how the corresponding WT mice were obtained, in the Materials and methods section of the revised manuscript.

3) Low and inconsistent numbers of replicates in the RNA-seq that reduce statistical power. This is especially problematic for attributing differential expression to loss of megadomains and superloop, because this analysis depended on comparing across multiple differential gene lists, which shrink with low statistical power (see only two Dxz4 ko replicates). This could be addressed by additional replicates from the single ko's, to get at least four across the board for liver, spleen and placenta.

We agree with the reviewer that having additional replicates from the SKO would increase the statistical power of the analysis in assigning differential expression to loss of *Firre* and megadomains and superloop. However, we only used the SKO to verify the direction of dysregulation observed in the DKO bodymap, where we had enough statistical power given that we used four biological replicates per group for this analysis.

4) Homozygous animals also lack Firre RNA (and possibly Dxz4-associated transcripts) expressed from the active X. The analysis here assumes that there are no confounding interactions between differential genes attributed to one group or another. A deeper analysis of the placental data in heterozygous F1 hybrid animals, where such interactions can be excluded, would be preferable and likely yield more insight towards identifying megadomain- or superloop-specific gene expression, if any.

Thank you very much for this suggestion. To address this point, we have performed an additional analysis to test whether the DKO deletion on either Xa or Xi might result in autosomal gene regulation in the placenta, a tissue where we can disentangle the functional role of *Firre* RNA on Xa from that of megadomain and superloop structures that only exist on the Xi. We detected only a few autosomal genes dysregulated in the DKO deletion on either Xa or Xi that were not changed in the SKO, suggesting that the mega-structures and the lncRNA *Firre* have no impact on autosomal gene regulation in the placenta (Supplementary file 1 sheet A-B). We have now highlighted this finding in the Results section of the revised manuscript, as follow: (“Notably, by using the same criteria we detected only a few autosomal genes dysregulated in the DKO that were not changed in the SKO, suggesting that the mega-structures and the lncRNA *Firre* have no impact on autosomal gene regulation in the placenta. (Supplementary file 1 sheet A-B).”)

In conclusion, while the authors present a worthwhile question and set of experiments, more careful analysis and interpretation is necessary to address functions of Firre and Dxz4 in XCI and development. At minimum, the placental RNA-seq should be repeated and fully analyzed after more than 5 backcrosses, with at least 4 replicates for each of the 4 genotypes presented.

We have now included a fourth biological replicate of the placenta DKO as indicated in point 2 of “Essential revision”. For the strain background/backcrossing issue please refer to “Essential revision” point 1.

Additional novelty could come from three interesting observations reported here: (1) enrichment for differential genes implicated in faithful chromosome segregation in mice lacking Firre expression, (2) Xist upregulation by 30% in Dxz4 single and double ko's (Figure 4—figure supplement 2B), and (3) shift in allelic ratio in the spleen of Dxz4 ko's (Figure 3—figure supplement 2B). Are there specific genes driving this or is the deletion skewing XCI?

a) It would be very interesting to follow up the *Firre* dependent enrichment of differential genes involved in chromosome structure and segregation, however we believe that this would be outside the scope of this study and will instead be followed up by a separate study.

b) *Xist* upregulation by 30% in *Dxz4* single and DKO is an interesting observation because it suggests that the *Dxz4* locus or the resulting megastructure is regulating *Xist* expression. Given that we don’t detect dysregulated genes on the X chromosome in the absence of *Dxz4*, our interpretation of this finding is that *Xist* upregulation occurs only from the inactive X chromosome, suggesting a role for maintaining X chromosome silencing. It would be very interesting to follow *Xist* expression in *Dxz4* deletions to test whether *Xist* upregulation correlates with aging, however, that is outside the scope of this current study.

c) In Figure 3—figure supplement 2B we show the allelic ratio of the E12.5 F1 brains (not the spleen) and found the expected XCI skewing ratios, a well-documented effect in female cells from crosses between Bl6 and CAST that results in the predominant inactivation of the Bl6 X chromosome. As pointed out by reviewer 2 and 3, the XCI skewing was even stronger in the *Dxz4* and DKO brains. To test whether the additional skewing is significant we performed a t-test on the *Xist* allelic ratios between WT (n=17) and each of the KO strains (n=6) and observed significant skewing (t-test, adjusted p-value = 0.0108) in the presence of the DKO deletion (Figure 3—figure supplement 2A right), suggesting that the Bl6 chromosome is further biased towards silencing in mice lacking both *Firre* and *Dxz4*. We have now highlighted this finding in the Results section of the revised manuscript, as follow: (“However, our DKO animals show significant skewing of the *Xist* allelic ratios (t-test, adjusted p-value = 0.0108) (Figure 3—figure supplement 2A right panel), suggesting that the Bl6 chromosome is further biased towards silencing in mice lacking both *Firre* and *Dxz4*.”)

Reviewer #3:1) The authors should perform Southern blotting and/or other assays to confirm that no autosomal off-target mutations were introduced when targeting the Firre and Dxz4 loci. PCR-screening alone as done at the moment (Figure 1—figure supplement 1) is not sufficient to rule that out especially considering the low number of back-crossings and the variability between replicates when looking at changes in autosomal gene expression (Figure 2D). This would be important especially as a main message of the paper is that there are Firre/Dxz4-dependent changes in autosomal gene expression.

Please refer to point 1 of “Essential revision”.

2) Figure 2: The different knockout alleles have been generated in mixed strain backgrounds (129, BALB/C, DBA) and only relatively few backcrosses seem to have been performed to pure B6 background (>2). Where did the wildtype mice come from, to which autosomal gene expression levels were compared in the adult expression bodymap? Were they of equivalent mixed strain background, or were wildtype controls of pure strain background used? It is important that controls of matched strain background were used throughout the study, in order to avoid differences in autosomal gene expression due to strain-background effects.

We agreed with the reviewer comment and have addressed this in “Essential revision” point 1-2 and in the response to reviewer 2 point 2.

3) Figure 3—figure supplement 2B and Figure 4—figure supplement 4C: Xist RNA seems to be upregulated dependent on Megadomain deletion (Figure 4—figure supplement 4C) and the allelic-ratio seems to skew random X-inactivation in the brain more towards the M Musculus (Dxz4-deleted) allele in Dxz4 and Firre/Dxz4 KO mice. The authors should check, if Xist skewing and allelic ratios of X-linked genes are significantly changed during random XCI in the Dxz4 and Firre/Dxz4 KOs.

Thank you very much for this observation; we have addressed this in reviewer 2 point 4c.

4) Figure 2D: The megadomain-specific upregulation of urinary cluster genes in liver seems to be very variable between replicates. It would be good to add a third replicate for delta Dxz4 as at the moment one replicate shows strong upregulation, while the second one doesn't. Also, in the Firre/Dxz4 double-mutants half of the replicates show an effect while others don't. How do the authors explain this discrepancy between replicates?

As noted by reviewer 2, the expression of major urinary protein genes was shown to be highly variable across individuals (PMID: 26973837) which explains the discrepancy across replicates.

[Editors' note: further revisions were requested prior to acceptance, as described below.]

Reviewer 1 is satisfied with the revisions (no comments attached). Reviewer 3 also feels that the paper is much improved but remains concerned about strain background. We know that it may be difficult to completely rule out strain background issues, and thus do not consider this to be an impediment to publication. However, reviewer 2 has a number of remaining concerns. In particular:1) Please include a supplementary table listing all RNA-seq samples as Rev2 suggested.

We listed all the RNA-seq samples in the supplementary table (sheet: J sample information) and updated the table legend.

2) Please present a Venn diagram listing the CAST/BL6 -common and specific escaping genes in placental RNA-seq.

We included a Venn diagram listing common and strain specific escaper genes in the placenta in Figure 3—figure supplement 2A.

3) Please add a p-value for that difference in median X-linked expression and Xist differences.

We have added the FDR-adjusted p-values for the X-linked genes.

4) Please report Dxz4 SKO & DKO sex ratios in Figure 1B.

As pointed out in the previous round of revision, the male-female ratio of the DKO progeny was not significantly altered (Figure 1B, 6 litters, 53 pups p=0.2712), suggesting that sex ratio skewing in the *Dxz4* SKO strain would be unlikely. For this reason, we kept *Dxz4* SKO line to a minimum and mainly used the strain to generate crosses with CAST for the allele-specific placenta analysis. The sex-ratio of the *Firre* SKO progeny is also normal (Lewandowski et al., in press) and, since the *Firre* locus is the main driver of the observed expression changes, we feel that assessing sex-ratio in the *Dxz4* SKO is outside the scope of this study.

The main biological question of this study was to investigate the in vivo role of the two Xlinked loci *Firre* and *Dxz4,* that provide the platform of the largest conserved chromatin structures in female mammals. We are confident that we addressed this question in the current manuscript and believe that the addition of the male-female ratios of the *Dxz4* SKO progeny will not change the conclusion of this study, bearing in mind that the ratios were not altered in the DKO.

5) Reviewer 2 also requested that you please re-analyze genome-wide differential expression without log2 FC cutoff in the placental RNA-seq data to distinguish the Firre locus from superloop and megadomain differentially expressed genes (if any) for Figure 4.

As suggested, we have re-analyzed genome-wide placental RNA-seq differential expression without log2FC cutoff. By comparing differentially expressed genes between the DKO and each of the SKO we find 3 overlapping genes between the DKO and *Firre* SKO on Xa (*Firre* RNA-specific), 45 genes between the DKO and *Firre* SKO on Xi (superloop-specific) and none between the DKO and *Dxz4* SKO on Xi (megadomainspecific). Of note, in all cases (with the exception of the deleted *Firre* gene), the mean log2FC between the DKO and SKO’s is close to zero (see Author response image 1). This result indicates that the mega-structures and the lncRNA *Firre* have no impact in gene regulation in the placenta as we observed in our original analysis (log2FC ≥1).

Given that these results are consistent with those from our original analysis, we did not include the new analysis in the revised manuscript.

6) Resolve the incongruence between the genes listed as commonly differentially expressed in Figure 4 and the gene lists in the supplementary file. Please see comments below for more details.

The genes listed in the supplementary file (Supplementary file 1 sheet G-H) and reported in the heatmap in Figure 4D (and Figure 4—figure supplement 2A) have been identified by differential expression analysis and represent genes dysregulated in at least one of the three KO strains (FDR ≤ 0.1) where the direction of the expression change in either one of the single KO agrees with that of the DKO (|log2FC| ≥ 1). The genes in Figure 4E (MA-plot and Venn diagram), instead, are the genes within the top 5 enriched dysregulated gene sets identified by the GSEA analysis in the DKO and *Firre* SKO spleen (Figure 4—figure supplement 2C).

To make clear that these are two different analyses we have added the following header to Figure 4E “GSEA identified chromosome structure and segregation gene sets”. We have also provided additional information in the method section and included the enriched GSEA gene sets and the corresponding DESeq2 output in Supplementary file 1 sheet I (genes ranked by lfcMLE) to allow for replication of the analysis. Given that all the information of figure 4E is now provided in the supplementary table, we removed the Venn diagram and instead added the spearman correlation of the fold changes.